# KIF18A's neck linker permits navigation of microtubule-bound obstacles within the mitotic spindle

Heidi LH Malaby, Dominique V Lessard, Christopher L Berger, Jason Stumpff

**KIF18A (kinesin-8) is required for mammalian mitotic chromosome alignment. KIF18A confines chromosome movement to the mitotic spindle equator by accumulating at the plus-ends of kinetochore microtubule bundles (K-fibers), where it functions to suppress K-fiber dynamics. It is not understood how the motor accumulates at K-fiber plus-ends, a difficult feat requiring the motor to navigate protein dense microtubule tracks. Our data indicate that KIF18A's relatively long neck linker is required for the motor's accumulation at K-fiber plus-ends. Shorter neck linker (sNL) variants of KIF18A display a deficiency in accumulation at the ends of K-fibers at the center of the spindle. Depletion of K-fiber–binding proteins reduces the KIF18A sNL localization defect, whereas their overexpression reduces wild-type KIF18A's ability to accumulate on this same K-fiber subset. Furthermore, single-molecule assays indicate that KIF18A sNL motors are less proficient in navigating microtubules coated with microtubule-associated proteins. Taken together, these results support a model in which KIF18A's neck linker length permits efficient navigation of obstacles to reach K-fiber ends during mitosis.**

## Introduction

Kinesin motor proteins are responsible for building and maintaining the mitotic spindle (Sawin et al, 1992; Tanenbaum et al, 2009), transporting, aligning, and orienting chromosomes at the metaphase plate (Levesque & Compton, 2001; Kapoor et al, 2006; Stumpff et al, 2008), and scaffolding the spindle midzone for cytokinesis (Kurasawa et al, 2004). These microtubule-dependent functions must be performed in the context of dense molecular environments because of the sheer number of microtubule-associated proteins that are found within mitotic spindles (Hughes et al, 2008).

The kinesin-8 motor protein KIF18A functions to confine chromosome movements around the metaphase plate and is required for proper chromosome alignment (Mayr et al, 2007; Stumpff et al, 2008). KIF18A accumulates at the ends of K-fibers, which are bundles

of ~15–20 microtubules collectively bound to kinetochore protein complexes assembled at centromeres (DeLuca & Musacchio, 2012). KIF18A accumulation on microtubule ends suppresses microtubule dynamics, dampening chromosome oscillations in metaphase (Stumpff et al, 2008, 2012; Du et al, 2010).

KIF18A is a dimeric, processive kinesin with two conserved globular motor domains that interact with microtubules through surface polar and positively charged residues (Gigant et al, 2013). Like all ATP-dependent kinesins, motor domain affinity for the microtubule surface correlates with nucleotide state. The neck linker, a short 14–17–amino-acid region between the end of the motor domain and the coiled-coil stalk, begins to dock to the motor domain upon a conformational shift induced by ATP binding and finishes docking upon hydrolysis of ATP to ADP and phosphate release (Milic et al, 2014). This creates a lever action to pull forward the ADP-bound trailing motor domain (Cross & McAinsh, 2014). Although this mechanical process is highly conserved among members of the kinesin family, small residue differences can alter the balance between processivity and microtubule affinity, determining the motor off rate (Cross & McAinsh, 2014). For example, increasing the neck linker length of kinesin-1 (*Drosophila* conventional, normally 14 residues) reduces its run length, whereas shortening the kinesin-2 neck linker (Kif3A, normally 17 residues) enhances its run length (Shastry & Hancock, 2010). In addition, the longer neck linker length of kinesin-2 provides the structural flexibility required to navigate around microtubule-bound obstacles (Telley et al, 2009; Hoeprich et al, 2014, 2017). These studies have all been performed in purified in vitro systems. The importance of neck linker flexibility for kinesin motility in cells is not understood.

Although a C-terminal microtubule-binding domain is necessary for KIF18A accumulation at K-fiber plus-ends in cells (Mayr et al, 2011; Stumpff et al, 2011; Weaver et al, 2011), it is not sufficient. A kinesin-1 chimera containing KIF18A's C-terminus does not accumulate on K-fiber ends, indicating there may be additional structural determinants of KIF18A's K-fiber end accumulation (Kim et al, 2014). Here, we determined if KIF18A's neck linker length, which is 17 residues, is important for its ability to accumulate at K-fiber ends. We engineered a panel of KIF18A short neck linker (sNL) constructs, which failed to accumulate at K-fiber ends at the center of the spindle. Furthermore, KIF18A sNL constructs were deficient

Department of Molecular Physiology and Biophysics, University of Vermont, Burlington, VT, USA

Correspondence: jstumpff@uvm.edu

in promoting chromosome alignment and progression through mitosis. Shortening the KIF18A neck linker creates a faster, less processive motor that is not as efficient at navigating obstacles on the microtubule. Consistent with an obstacle navigation role for KIF18A's neck linker in cells, reducing the microtubule-bundling protein HURP or the kinetochore mesh component TACC3 allows KIF18A sNL to accumulate at the plus-ends of central K-fibers, whereas increased hepatoma-upregulated protein (HURP) expression inhibits accumulation of the wild-type motor. Taken together, this study supports a model in which KIF18A's long neck linker tunes the motor for navigation of the densely populated K-fiber surface.

## Results

### sNL variants of KIF18A do not accumulate at the ends of K-fibers in the center of the spindle

To investigate the importance of KIF18A neck linker length for its ability to accumulate at K-fiber plus-ends in mitotic cells, we created a panel of sNL KIF18A variants (Fig 1A). To determine where to make these deletions, we used a combination of coiled-coil prediction algorithms (COILS and LOGICOIL) and sequence alignments with the previously identified neck linkers of KIF5B and KIF3A (Shastry & Hancock, 2010). We concluded that KIF18A's probable neck linker is 17–amino-acid long, including residues 353–369. Because coiled-coil prediction algorithms are inaccurate

at pinpointing the beginning of coiled-coil domains and altering the sequence of the kinesin neck linker can affect the coiled-coil (Phillips et al, 2016), we created a series of four sNL variants, taking care to keep the highly conserved N362 in frame (sNL0, sNL1, sNL2, and sNL3; Fig 1A).

To test the localization of these constructs in mitotic cells, we treated HeLa cells with KIF18A siRNA before transfection with each GFP-KIF18A sNL variant, which also contained silent mutations in the siRNA-targeted region. Cells were arrested in metaphase by treatment with MG132, fixed, and stained with a GFP antibody. Strikingly, the localizations of GFP-KIF18A sNL1, sNL2, and sNL3 in mitotic cells were similar and did not accumulate on K-fibers at the center of the spindle (Fig 1B, white arrows), but they did accumulate on K-fibers near the spindle periphery (Fig 1B, blue arrows and Fig S1). Aligned line scans from many cells indicated that the lack of accumulation observed at plus-ends of K-fibers in the center of the spindle was highly reproducible for these GFP-KIF18A sNL variants (Fig 1C). By contrast, GFP-KIF18A sNL0 displayed a uniform spindle distribution with no K-fiber end accumulation (Fig S2A). We predict that loss of H369 in this truncation mutant may disrupt formation of the coiled-coil region that is C-terminal to the neck linker (Phillips et al, 2016). Consistent with this interpretation, we found that treatment of cells with taxol for 10 min before fixation, which stabilizes K-fibers and promotes KIF18A end accumulation (Kim et al, 2014), allowed some GFP-KIF18A sNL0 to localize at K-fiber plus-ends in the majority of cells (~70%) only when endogenous KIF18A was present. However, this localization was absent in most cells (>80%) when endogenous KIF18A was knocked down (Fig S2B).

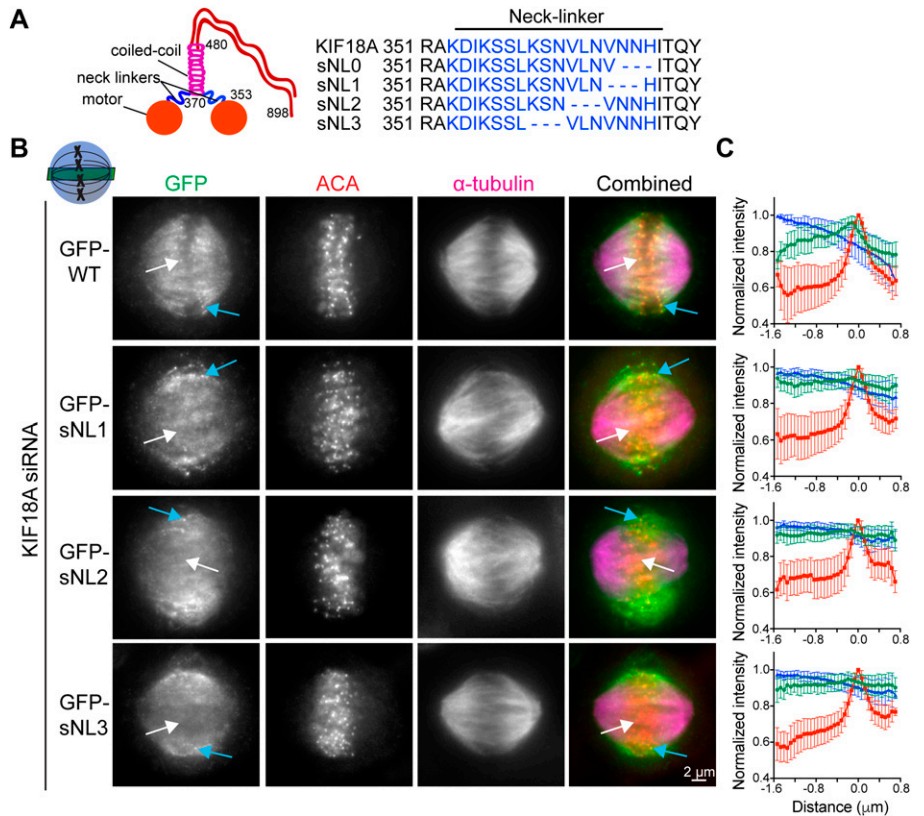

**Figure 1. KIF18A sNL variants do not accumulate on K-fibers in the center of the spindle.**
**(A)** KIF18A putative structure and predicted neck linker region. Cartoon depiction of known KIF18A structural regions with residue numbers and predicted neck linker residues (blue) of KIF18A and sNL variants used in this study. **(B)** Representative metaphase-arrested cells expressing GFP-WT KIF18A or GFP-sNL1-3 variants. Shown is one z-slice through the middle of the spindle. White arrows point to a central K-fiber in each cell and blue arrows point to a peripheral K-fiber. **(C)** Quantification of KIF18A accumulation on central K-fiber ends by line scan analysis. Individual lines were normalized and aligned to peak ACA intensity. Symbols show normalized intensity average; error bars are SD. Red, ACA; blue, α-tubulin; and green, KIF18A variant. $P < 0.0001$ by F test comparing slopes of linear regressions for WT versus any sNL variant. Data obtained from two independent experiments with the following cell and line numbers: GFP-WT (13 cells, 40 lines), GFP-sNL1 (17 cells, 49 lines), GFP-sNL2 (15 cells, 51 lines), and GFP-sNL3 (20 cells, 62 lines). Source data are available for this figure.

Overall, these results indicate that (1) KIF18A sNL1-3 are active motors and (2) a longer neck linker is necessary for accumulation at K-fiber ends in the center of the mitotic spindle.

## KIF18A sNL variants are deficient in chromosome alignment and cause a mitotic delay

In metaphase cells lacking KIF18A, chromosomes and kinetochores are broadly distributed within the spindle (Mayr et al, 2007; Stumpff et al, 2008). To determine if the KIF18A sNL mutants were competent to align mitotic chromosomes, we knocked down endogenous KIF18A in HeLa cells and transfected each GFP-KIF18A sNL variant. We confirmed knockdown of endogenous KIF18A and similar transient expression of GFP-KIF18A by Western blot (Fig 2A). For comparison, we transfected KIF18A knockdown cells with GFP-WT KIF18A as a positive control and GFP alone as a negative control. Cells were arrested in metaphase, fixed, and stained for kinetochores, GFP, and γ-tubulin to label spindle poles (Fig 2B). To quantify alignment, we measured the distribution of kinetochore (ACA) immunofluorescence along the spindle axis and determined the full width at half maximum (FWHM) of this distribution

(Kim et al, 2014; Stumpff et al, 2012). In agreement with previous work, chromosome alignment is significantly improved by expression of GFP-WT KIF18A but remains disrupted in cells expressing GFP alone (Fig 2C) (Kim et al, 2014). The GFP-sNL 1–3 KIF18A variants produced intermediate chromosome alignment phenotypes, where chromosomes off the metaphase plate were often in the central region of the spindle where the motor did not accumulate (Fig 2B, white arrows). Chromosome alignment was similarly defective in cells expressing GFP-KIF18A sNL0 or GFP alone (Fig S2C). KIF18A is also required to maintain spindle length (SL) (Mayr et al, 2007). Quantification of pole-to-pole distances revealed that KIF18A sNL variants displayed intermediate SL (Fig 2C). Because KIF18A sNL1-3 variants displayed similar localization and effects on alignment, we focused the rest of our studies on KIF18A sNL1. The sNL0 mutant was not pursued further because the phenotypes of sNL0-expressing cells were significantly different from those seen in cells expressing the other three shortened neck linker constructs.

Loss of KIF18A function in HeLa cells also leads to an extended mitotic delay that is dependent on the spindle assembly checkpoint (Mayr et al, 2007; Zhu et al, 2005). Control HeLa cells progress

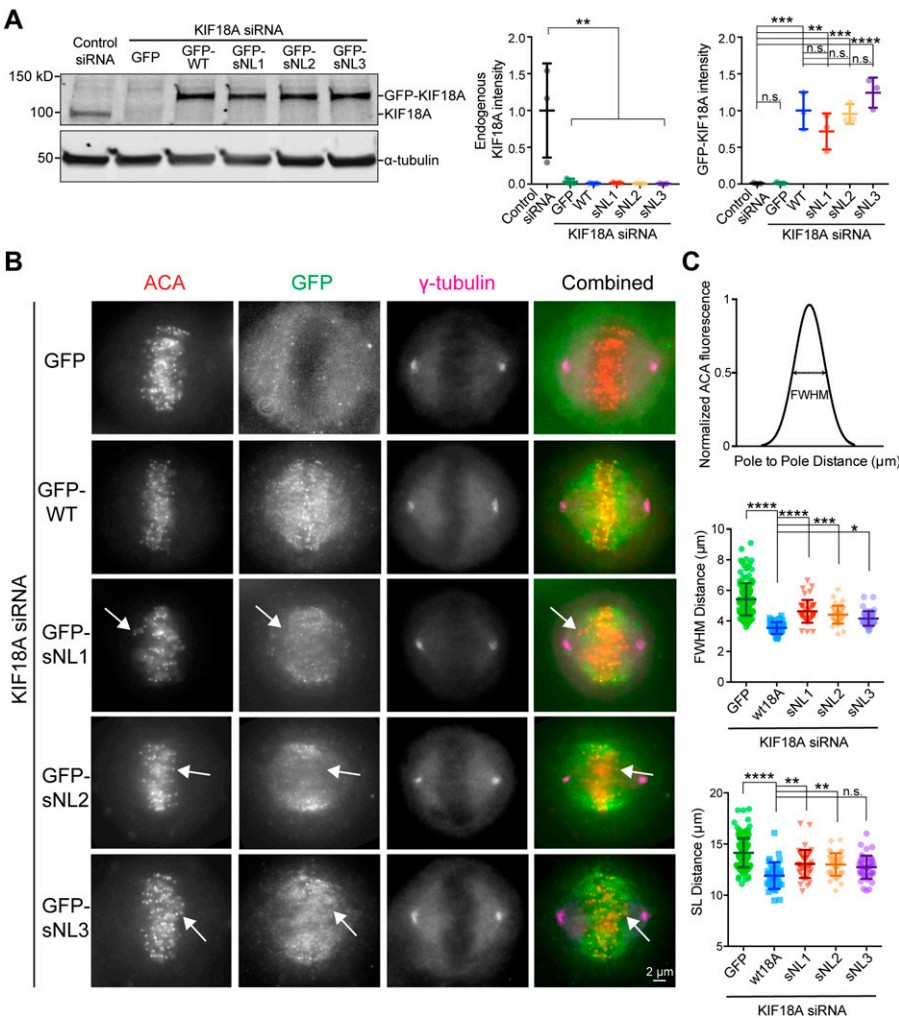

**Figure 2. KIF18A sNL variants are deficient in chromosome alignment.**
**(A)** Western blot quantification of knockdown and overexpression conditions used in this study. Top blot, KIF18A; bottom blot, α-tubulin. Quantification to right is from three independent assays. **, adjusted $P <$ 0.01; ***, adjusted $P <$ 0.001; ****, adjusted $P <$ 0.001; n.s., not significant, with 95% confidence interval by one-way ANOVA with Tukey's multiple comparisons test. **(B)** Representative images from metaphase-arrested cells treated with KIF18A siRNAs and transfected with GFP alone, GFP-WT KIF18A, or GFP-sNL1-3 variants. White arrows mark a chromosome pair off the metaphase plate in the center of the spindle where GFP-sNL variant is not accumulating. Images are one z-plane through the middle of the cell where both poles are in focus. **(C)** Graphical explanation of ACA fluorescence signal intensity FWHM measurement (top); FWHM distance (middle); and SL distance quantifications for all conditions (bottom). Bars are mean ± SD. ****, adjusted $P <$ 0.0001; ***, adjusted $P =$ 0.0001; *, adjusted $P =$ 0.0119, with 95% confidence interval by one-way ANOVA with Tukey's multiple comparisons test. Data obtained from three independent experiments with the following cell numbers: GFP only (141), GFP-WT KIF18A (35), GFP-sNL1 (41), GFP-sNL2 (38), and GFP-sNL3 (43).
Source data are available for this figure.

through mitosis in ~40 min, whereas cells deficient in KIF18A can be arrested for hours (Hafner et al, 2014; Zhu et al, 2005). Endogenous KIF18A was knocked down in HeLa cells, and the cells were transfected with GFP-WT KIF18A, GFP-sNL1, or GFP alone (Fig 3A). The cells were analyzed by time-lapse microscopy and the time from nuclear envelope breakdown to anaphase onset was determined for each GFP-expressing cell that divided completely. GFP-sNL1 KIF18A displayed a ~20 min mitotic delay compared with GFP-WT KIF18A (Fig 3B), which has been shown to rescue mitotic timing to ~40 min in HeLa cells (Hafner et al, 2014). Taken together, these findings show that KIF18A's long neck linker is required for complete chromosome alignment and timely progression through mitosis.

### KIF18A sNL1 is faster than WT KIF18A but has a shorter run length

Neck linker length is an important determinant of run length and velocity for kinesin-1 and kinesin-2 motors. For example, shortening the 17 residue kinesin-2 neck linker increases both run length and velocity (Shastry & Hancock, 2010). To determine what effect shortening the KIF18A neck linker has on its motile properties, we compared purified KIF18A WT and sNL1 in single-molecule assays. Because we wanted to observe only the neck linker's influence on velocity and run length, we used truncated KIF18A constructs (GFP-KIF18A$^{1-480}$) that lack the C-terminal microtubule-binding domain. The movements of individual GFP-tagged motors were visualized on rhodamine-labeled microtubules via total internal reflection fluorescence (TIRF) microscopy (Fig 4A). The velocity and run length of GFP-KIF18A$^{1-480}$ WT was comparable with those measured for other truncated KIF18A constructs (Mayr et al, 2011; Stumpff et al, 2011). By contrast, GFP-KIF18A$^{1-480}$ sNL1 was almost twice as fast as the WT motor but moved about one-third the distance (Fig 4B and C and Table S1). However, this decreased run length alone would not explain why KIF18A sNL variants are able to accumulate at the ends of peripheral but not central K-fibers, especially because peripheral

K-fibers, assuming relative continuity, are predicted to be slightly longer because of the geometry of the spindle.

### The motility of KIF18A sNL1 is more hindered in the presence of obstacles than WT KIF18A

The structural flexibility imparted by the long neck linker of kinesin-2 permits navigation around obstacles on microtubules (Hoeprich et al, 2014, 2017). To determine how KIF18A's motility is affected by the presence of microtubule-associated proteins, we compared the movements of single GFP-KIF18A$^{1-480}$ molecules in the presence and absence of tau or an immobile, rigor kinesin mutant. The microtubule-associated protein tau binds both statically and diffusely to the microtubule surface (Hinrichs et al, 2012; McVicker et al, 2014), and its presence reduces the run length of kinesin-1 (Dixit et al, 2008; McVicker et al, 2011; Vershinin et al, 2007). We found that a concentration of 1:40 tau to tubulin dimer reduced the run length of GFP-KIF18A$^{1-480}$ WT by ~10%. However, tau's effect on run length was significantly larger (over 50%) for GFP-KIF18A$^{1-480}$ sNL1 (Fig 4D–F and Table S1). Tau did not alter the velocity of either motor. Consistent results were obtained using rigor kinesin as an obstacle at a concentration of 1:12.5 rigor kinesin to tubulin dimer, which also limits the movement of kinesin-1 motors (Hoeprich et al, 2017). The presence of rigor kinesin reduced the run length of GFP-KIF18A$^{1-480}$ WT by ~12% but decreased the run length of GFP-KIF18A$^{1-480}$ sNL1 twice as much (~23%) (Fig S3A–D and Table S1). Taken together, these findings indicate WT KIF18A is capable of obstacle navigation on microtubules, and shortening KIF18A's neck linker significantly compromises this activity.

### Localization of KIF18A sNL1 at the ends of K-fibers is reduced by mitotic microtubule-associated proteins

Our single-molecule studies suggest that the localization and functional defects observed for KIF18A sNL variants in mitotic cells

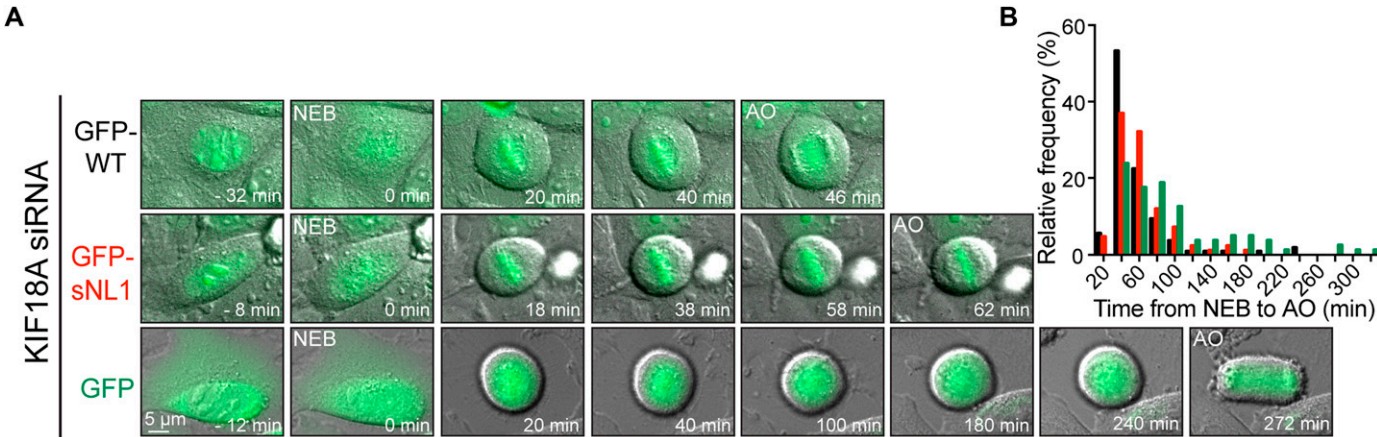

**Figure 3. KIF18A sNL variant fails to promote proper mitotic progression.**
**(A)** Time course DIC and GFP overlay images from live cells treated with KIF18A siRNAs and then transfected with GFP-WT KIF18A, GFP-sNL1, or GFP alone. Times are normalized to NEB. **(B)** Histogram distributions of the time observed from NEB to AO for all cells with GFP expression that finished division by the end of imaging. For chi-square analysis, all conditions were subdivided into 20-min bins and then grouped into three timeframes: <40 min, 60–80 min, and >100 min; n = individual cells. GFP-WT KIF18A (expected) to GFP-sNL1 (observed): $P = 0.0247$; GFP-WT KIF18A (expected) to GFP alone: $P < 0.0001$; and GFP alone (expected) to GFP-sNL1 (observed): $P = 0.0006$. Data obtained from three independent experiments with the following cell numbers: GFP-WT KIF18A (107), GFP-sNL1 (84), and GFP alone (80). AO, anaphase onset; NEB, nuclear envelope breakdown.

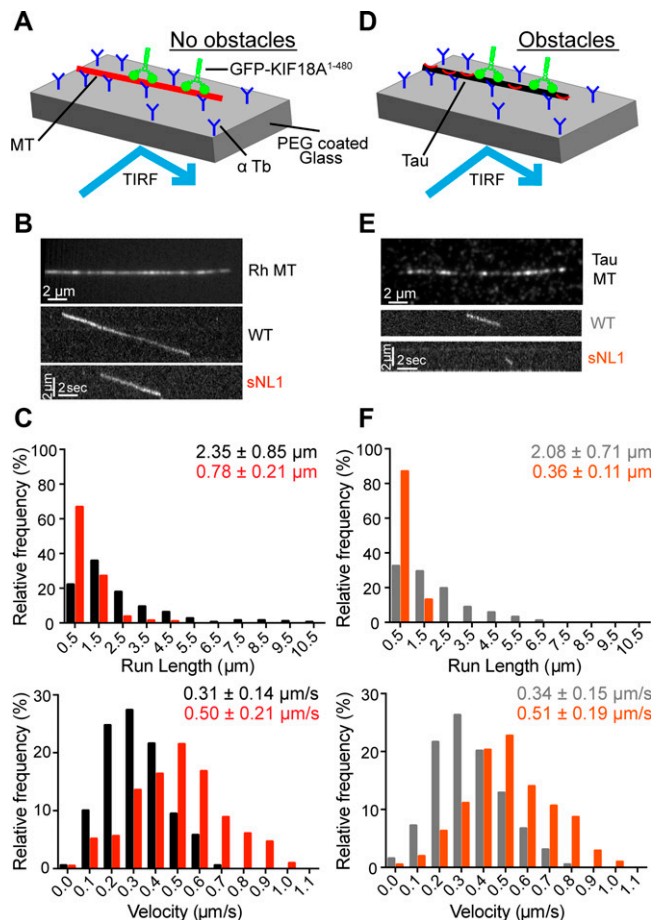

**Figure 4. KIF18A sNL variant is less processive but faster than WT KIF18A and more hindered by obstacles.**
**(A)** Cartoon depiction of single-molecule TIRF assay setup without obstacles on microtubules. **(B)** Picture of single rhodamine (Rh)-labeled MT in single-molecule TIRF (top); representative kymographs of GFP-KIF18A$^{1-480}$ WT (WT) and GFP-KIF18A$^{1-480}$ sNL1 (sNL1) processive events. **(C)** Histogram distributions of run lengths (*top*) and velocities (*bottom*) of processive events observed. *P* < 0.0001 for both run length and velocity comparing WT and sNL1 by unpaired *t* test. **(D)** Cartoon depiction of single-molecule TIRF assay set up with Alexa 647 3RS tau as an obstacle on MT. **(E)** Picture of single Alexa 647 3RS tau–coated MT (top); representative kymographs of GFP-KIF18A$^{1-480}$ WT (WT) and GFP-KIF18A$^{1-480}$ sNL1 (sNL1) processive events on tau-coated MT (bottom). **(F)** Histogram distributions of run lengths (top) and velocities (bottom) of processive events observed on tau-coated MT. Statistical significance comparing no tau versus tau by unpaired *t* test: run length, WT *P* = 0.0006; sNL1 *P* < 0.0001; velocity, WT *P* = 0.0391; and sNL1 *P* = 0.6170. Data obtained from two independent protein preps and 16 individual TIRF assays with the following total processive events: no tau: WT (190), sNL1 (214); with tau: WT (194), sNL1 (207). MT, microtubules.
Source data are available for this figure.

could be caused by an inability of the motor to navigate around microtubule-associated proteins to reach the ends of K-fibers. The microtubule-stabilizing protein HURP regulates KIF18A's localization and function in mitotic cells (Ye et al, 2011). HURP localizes as a gradient on K-fibers with a concentration near the plus-ends because of Ran regulation (Koffa et al, 2006; Sillje et al, 2006). In HURP-depleted cells, KIF18A displays a tighter accumulation at K-fiber plus-ends, whereas overexpression of HURP's microtubule-binding domain disrupts KIF18A's localization and function. Interestingly,

these latter effects can be rescued by overexpression of KIF18A (Ye et al, 2011). These results are consistent with HURP functioning as an obstacle or filter for motor movement to the K-fiber plus-end.

To test this idea, we treated HeLa cells with siRNAs against KIF18A or KIF18A and HURP, then transfected GFP-WT KIF18A or GFP-sNL1. The cells were arrested in metaphase, fixed, and stained for GFP, endogenous HURP, and kinetochores (ACA) (Fig 5A). To quantify KIF18A accumulation changes at K-fiber ends in the center of the spindle, we generated normalized, aligned line scans of GFP fluorescence as in Fig 1 and determined the area under the curve for K-fiber regions close to (tip) and farther from (lattice) the kinetochore. We computed the tip to lattice ratio of KIF18A fluorescence as a metric for its plus-end accumulation (Fig 5C). After HURP knockdown, GFP-WT KIF18A showed a slight decrease in central K-fiber accumulation, which we attribute to less microtubule tracks present per K-fiber bundle upon HURP knockdown (Fig S4) (Wong & Fang, 2006). Despite the decrease in K-fiber attachment and stability caused by HURP-depletion, the accumulation of GFP-sNL1 KIF18A significantly increased at the ends of K-fibers in the center of the spindle (Fig 5C, middle). Similar GFP-WT and GFP-sNL1 KIF18A accumulation alterations were obtained by knocking down TACC3 (Fig S5), another microtubule-bound protein implicated in bridging K-fibers in complex with ch-TOG and clathrin (Booth et al, 2011). Moreover, accumulation of GFP-WT KIF18A could be suppressed by overexpression of a putative obstacle. When we knocked down endogenous KIF18A and then co-expressed GFP-HURP, which led to increased tubulin immunofluorescence in K-fibers (Fig S4), with either mCherry-tagged KIF18A WT or sNL1 (Fig 5B), WT KIF18A accumulation was reduced at the ends of K-fibers at the center of the spindle (Fig 5C, bottom).

We also questioned how KIF18A sNL variants could accumulate on peripheral but not centrally located K-fibers within the spindle, as both sets of K-fibers appear to contain HURP. However, closer examination of HURP and KIF18A localization indicates that both WT and KIF18A sNL localize to zones of low HURP on peripheral K-fibers, suggesting that KIF18A sNL1 is able to accumulate on peripheral K-fibers because HURP, and possibly other microtubule-associated proteins, is not uniformly present on these tracks (Fig 6A and source data). Taken together, these results indicate that the presence of microtubule-bound obstacles on K-fibers reduces KIF18A's ability to accumulate at plus-ends and that this effect is exacerbated by shortening the KIF18A neck linker.

## Discussion

Our work supports a model in which KIF18A must be agile during its translocation along K-fibers to navigate microtubule-associated proteins such as HURP and TACC3 and accumulate at plus-ends, where it controls chromosome movements (Fig 6B). KIF18A's relatively long 17–amino-acid neck linker is an important molecular component underlying the motor's navigation activity. Based on studies of kinesin-1 and kinesin-2, we predict that KIF18A's neck linker region provides structural flexibility, allowing the motor to step from one protofilament to the next as it moves toward microtubule plus-ends (Hoeprich et al, 2017). In support of this idea,

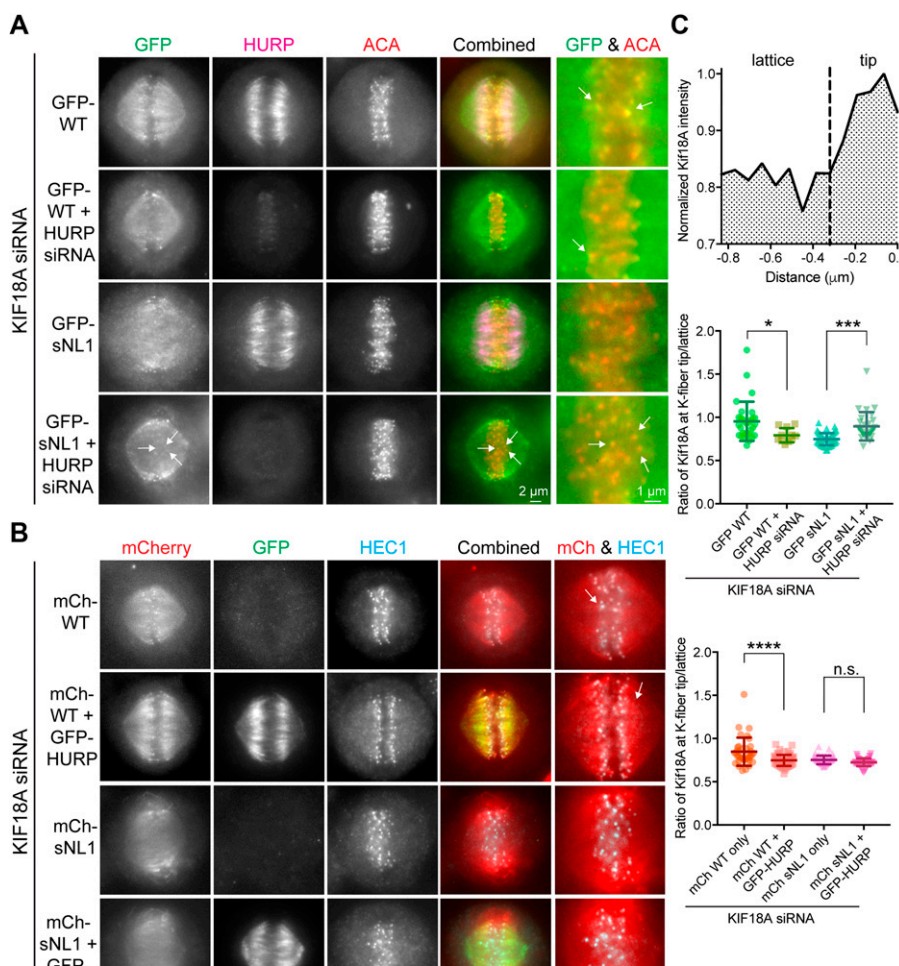

**Figure 5. HURP is an obstacle for KIF18A on K-fibers in metaphase cells.**
**(A)** Metaphase-arrested cells treated with KIF18A siRNA (and HURP siRNAs in marked conditions) and transfected with GFP-WT KIF18A or GFP-sNL1. Arrows indicate GFP-KIF18A accumulation near kinetochores (ACA, red). **(B)** Metaphase-arrested cells treated with KIF18A siRNA and transfected with mCherry-WT KIF18A, mCherry-sNL1, and/or GFP-HURP. Arrows mark the regions of greatest accumulation of mCherry-WT KIF18A. Shown in (A) and (B) is one z-slice through the middle of the spindle. **(C)** Quantification of KIF18A accumulation at central K-fiber ends under the conditions mentioned earlier. Line scans were performed, normalized, and aligned as in Fig 1, then each line was divided at the dotted line (−0.32 $\mu$m) and the area under the curve calculated for each segment (top). The ratio of tip (0 to −0.32 $\mu$m from kinetochore) to lattice (−0.32 to −0.768 $\mu$m) Kif18A fluorescent intensity is reported in the following graphs. Quantification of GFP-WT and GFP-sNL1 with HURP knockdown (middle). Bars are mean ± SD. *, adjusted $P$ = 0.0135; ***, adjusted $P$ = 0.0002 with 95% confidence interval by one-way ANOVA with Tukey's multiple comparisons test. Data obtained from two independent experiments with the following cell and line numbers: GFP-WT (12 cells, 31 lines), GFP-WT + HURP siRNA (7 cells, 11 lines), and GFP-sNL1 (14 cells, 50 lines). GFP-sNL1 + HURP siRNA (12 cells, 29 lines). Quantification of mCherry-WT and mCherry-sNL1 with GFP-HURP overexpression (bottom). ****, adjusted $P$ < 0.001; n.s., not significant (adjusted $P$ = 0.2061) with 95% confidence interval by one-way ANOVA with Tukey's multiple comparisons test. Data obtained from two independent experiments with the following cell numbers: mCh-WT (13 cells, 37 lines), mCh-WT + HURP-GFP (16 cells, 63 lines), mCh-sNL1 (12 cells, 48 lines), and mCh-sNL1 + GFP-HURP (16 cells, 51 lines). Source data are available for this figure.

the budding yeast ortholog of KIF18A, Kip3p, which shares a highly conserved neck linker region, is capable of switching protofilaments during movement (Bormuth et al, 2012; Bugiel et al, 2015). However, a physiological role for this type of side-stepping has not been determined for any kinesin. We propose that this lateral movement is necessary for KIF18A's mitotic functions based on our observations that sNL variants of KIF18A (1) display hindered motility in the presence of microtubule-bound obstacles in vitro, similar to kinesin-1 and kinesin-2 motors with sNLs (Telley et al, 2009; Shastry & Hancock, 2010; Hoeprich et al, 2014, 2017), (2) are not able to accumulate properly on K-fibers in mitotic cells unless microtubule-associated proteins are depleted, and (3) fail to align chromosomes or promote proper progression through mitosis in the absence of endogenous KIF18A.

Our model that microtubule-associated proteins act as obstacles for KIF18A is also consistent with previously published data showing that overexpression of HURP or the microtubule-binding domain of HURP inhibits KIF18A accumulation at K-fiber plus-ends (Ye et al, 2011). Increases or decreases in HURP expression directly affect the stability of microtubules in K-fibers, a finding that is true for other microtubule-bundling proteins, including TACC3 (Booth

et al, 2011; Nixon et al, 2015). HURP or TACC3 knockdown reduces K-fiber stability and WT KIF18A accumulation at K-fiber plus-ends. By contrast, KIF18A sNL1 accumulation at K-fiber plus-ends increases following HURP or TACC3 knockdown. These results are consistent with obstacles being removed and the potential of KIF18A sNL1 to reach the ends of less stable microtubules faster than WT KIF18A based on its increased velocity. On the other hand, HURP overexpression in cells stabilizes K-fibers but also reduces WT KIF18A accumulation at plus-ends. These results suggest that a balance of KIF18A and mitotic microtubule-associated proteins like HURP must be maintained to ensure that chromosome movements are properly controlled and spindle stability is maintained.

Our single-molecule studies indicate that KIF18A sNL1 is twice as fast as WT KIF18A but only travels a third of the distance. By contrast, shortening KIF3A's neck linker from 17 to 14 residues resulted in a faster motor that traveled longer distances (Hoeprich et al, 2014) and lengthening kinesin-1's neck linker from 14 to 17 residues reduced both the run length and velocity of the motor (Shastry & Hancock, 2010). Thus, altering neck linker length differentially affects the processivity of KIF18A compared with kinesin-1 and kinesin-2. Although we do not yet understand the molecular basis

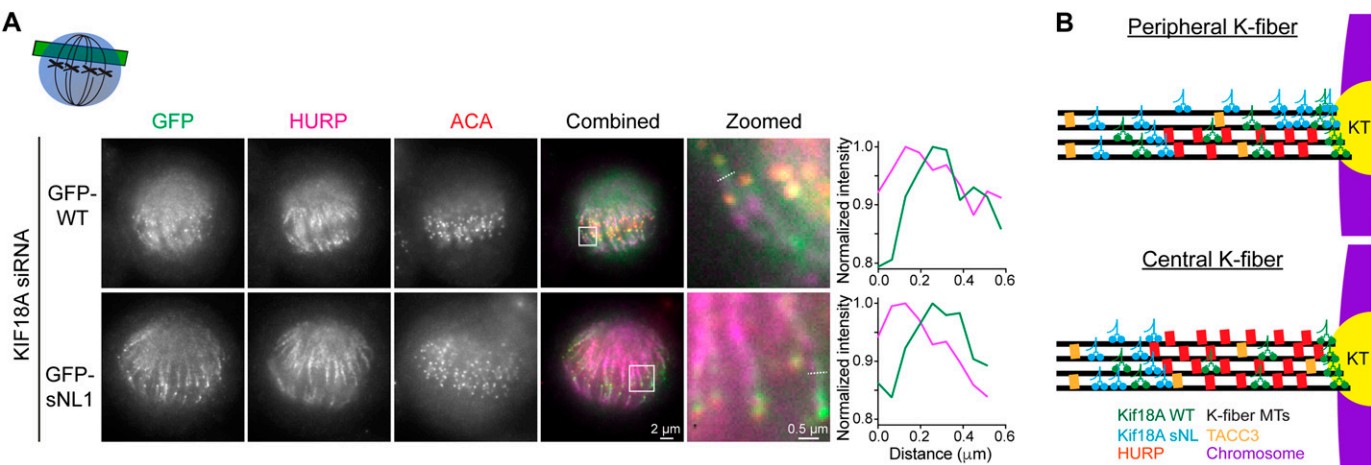

**Figure 6. HURP and KIF18A do not co-localize on peripheral K-fibers.**
**(A)** Shown are example metaphase-arrested cells treated with KIF18A siRNA and transfected with GFP-WT or GFP-sNL1 before fixation and staining with anti-HURP antibodies. Single z-slices of off-axis cells are shown to highlight peripheral K-fibers. White box denotes zoomed region. White dashed lines in zoomed images show line scan locations. Line scans are representative of 19 scans from 8 GFP-WT cells and 21 scans from 9 GFP-sNL1 cells. Green, GFP-KIF18A variant; pink, HURP. See source data for additional line scan examples. **(B)** Model for the role of KIF18A's neck linker in K-fiber end accumulation. KIF18A WT is ideally suited to navigate obstacles, such as HURP or TACC3, on K-fibers. Conversely, shortening KIF18A's neck linker reduces this navigation ability and limits accumulation to the outside of peripheral K-fibers. KT, kinetochore protein complex.
Source data are available for this figure.

of these differences, they may be related to unique loops in kinesin-8 family motor domains that allow extended dwell times at microtubule ends and promote depolymerization of microtubules by Kip3p (Kim et al, 2014; Arellano-Santoyo et al, 2017). Microtubule structure at a growing end is curved (Brouhard & Rice, 2014), unlike the rigid straight tracks within the lattice, and many proteins have a higher affinity for the growing tip (Honnappa et al, 2009; Bechstedt et al, 2014). Therefore, it is possible that KIF18A's affinity for the microtubule lattice is reduced compared with kinesin-1 or 2 motors, a trait that may be exacerbated by KIF18A sNL1's faster velocity. Also of note, our single-molecule studies were performed with C-terminal truncated forms of KIF18A to observe the influence of neck linker length without the additional tail microtubule-binding site. However, the full-length protein used in the cellular studies does have this extra tether, which has been shown to increase the processivity of KIF18A (Stumpff et al, 2011). The fact that KIF18A sNL variants are able to accumulate at the ends of peripheral K-fibers suggests that the decrease in run length observed for the truncated sNL motor in vitro does not fully explain its cellular localization phenotype. A general processivity defect would be expected to reduce KIF18A's ability to reach the end of all K-fibers and might preferentially reduce accumulation at the ends of longer microtubules within peripheral K-fibers. By contrast, sNL mutants display the opposite localization pattern, preferentially accumulating at peripheral K-fibers.

Although our data indicate that KIF18A's neck linker length is important for navigating K-fibers, there are certainly additional obstacles that KIF18A must contend with on K-fibers beyond HURP and TACC3. Eg5 and KIF15 are kinesin motors that stabilize the spindle and occupy the same microtubule-binding site as KIF18A (Tanenbaum et al, 2009). The dynein/dynactin complex, which also shares an overlapping binding site with kinesins (Mizuno et al, 2004), transports checkpoint-silencing proteins such as MAD2 and BUBR1

toward the spindle poles along K-fibers (Silva et al, 2014). Near the K-fiber plus-ends, the kinetochore protein complex is its own gauntlet of dense obstacles (Cheeseman & Desai, 2008). K-fibers are undoubtedly crowded tracks, but here we provide evidence that KIF18A is uniquely adapted to this environment to reach and accumulate at K-fiber ends. Importantly, this localization is required for the timely alignment and segregation of mitotic chromosomes.

# Materials and Methods

### Plasmids and siRNAs

GFP-KIF18A sNL variants were constructed by Gateway Cloning technology (Thermo Fisher Scientific) and QuikChange site-directed mutagenesis (Agilent) into a pCMV-eEGP-N1 backbone containing the full-length WT KIF18A sequence with N-terminal GFP tag and C-terminal His tag in frame. Specific deletions for each construct are shown in Fig 1A. For protein purification, WT KIF18A and sNL1 were recombined from a pCR8/GW/TOPO entry vector into a pET-eGFP destination vector with N-terminal GFP tag and C-terminal FLAG tag with an LR reaction (Thermo Fisher Scientific). pmaxGFP (Lonza) used for GFP-only controls. Scrambled control siRNA obtained from Thermo Fisher Scientific (Silencer Negative Control #2). KIF18A siRNA antisense sequence (Stumpff et al, 2008, 2012; Kim et al, 2014): 5'-GCUGGAUUUCAUAAAGUGG-3' (Ambion). HURP siRNA antisense sequence: 5'-AAGGAGUCCAGGUGUAACUGG-3' (QIAGEN). TACC3 siRNA antisense sequence: 5'UGCUUUCCUCAAGGGAGGC-3' (Ambion).

### Cell culture and transfections

HeLa cells were cultured at 37°C with 5% $CO_2$ in MEM-$\alpha$ medium (Life Technologies) containing 10% FBS plus 1% penicillin/streptomycin

(Gibco). For siRNA transfections in a 24-well format, approximately 80,000 cells in 500 $\mu$l MEM-$\alpha$ medium were treated with 35 pmol siRNA and 1.5 $\mu$l RNAiMax (Life Technologies), preincubated for 20 min in 62.5 $\mu$l OptiMeM (Life Technologies). The cells were treated with siRNAs for 48 h before fixing or imaging. 24 h before fixation or imaging, the medium was changed and plasmid transfections were conducted similarly, but with 375 ng KIF18A or HURP plasmid DNA and 2 $\mu$l LTX (Life Technologies).

## Cell fixation and immunofluorescence

For fixation, HeLa cells were treated with 20 $\mu$M MG132 (Selleck Chemicals) for 1–2 h before fixation. The cells were fixed on coverslips in –20°C methanol (Thermo Fisher Scientific) plus 1% para-formaldehyde (Electron Microscopy Sciences) for 10 min on ice, dunked briefly, and then washed 3× for 5 min each in TBS (150 mM NaCl and 50 mM Tris base, pH 7.4). Coverslips were blocked with 20% goat serum in antibody-diluting buffer (Abdil: TBS, pH 7.4, 1% BSA, 0.1% Triton-X, and 0.1% sodium azide) for 1 h at room temperature before the primary antibodies diluted in Abdil were added for 1 h at room temperature: mouse anti–$\alpha$-tubulin at 1 $\mu$g/ml (Sigma-Aldrich), mouse anti–$\gamma$-tubulin at 1 $\mu$g/ml (Sigma-Aldrich), rabbit anti-GFP at 4 $\mu$g/ml (Molecular Probes), rabbit anti-HURP at 2 $\mu$g/ml (Bethyl Laboratories), and mouse anti-TACC3 at 0.4 $\mu$g/ml (Santa Cruz). Primary antibodies against human anti-centromere protein (ACA) at 2.5 $\mu$g/ml (Antibodies Incorporated) and mouse anti-HEC1 at 0.5 $\mu$g/ml (GeneTex) were used overnight at 4°C. All mCherry images are direct measurement of mCherry fluorescence. Incubation with 1 $\mu$g/ml goat secondary antibodies against mouse, rabbit, or human IgG conjugated to Alex Fluor 488, 594, or 647 (Molecular Probes) occurred for 1 h at room temperature. 2× TBS 5 min washes were performed between rounds of primary and secondary antibody incubations, finishing with 3× TBS washes before mounting with Prolong Gold antifade mounting medium with DAPI (Molecular Probes).

## Western blot analysis

For comparison of exogenous KIF18A expressions, approximately 350,000 cells were plated into a six-well dish and transfected as described earlier (scaled siRNA: 150 pmol siRNA + 6 $\mu$l RNAiMax in 250 $\mu$L OptiMeM; scaled plasmid: 1.5 $\mu$g plasmid + 8 $\mu$l LTX in 250 $\mu$l OptiMeM). The cells were washed once in PBS (Thermo Fisher Scientific) and lysed in 200 $\mu$l PHEM lysis buffer (60 mM PIPES, 10 mM EGTA, 4 mM MgCl$_2$, and 25 mM Hepes, pH 6.9) with 1% Triton-X and Halt protease and phosphatase inhibitors (Thermo Fisher Scientific) by scraping the plate and incubating on ice for 10 min. The lysates were then spun at maximum speed for 10 min and 180 $\mu$l supernatant added to 60 $\mu$l 4× Laemmli buffer prepared with $\beta$ME (Bio-Rad). The samples were run on precast 4–15% gradient gels (Bio-Rad), transferred to low-fluorescence polyvinylidene fluoride membrane (Bio-Rad), blocked for 1 h in 1:1 Odyssey Blocking Buffer (Li-Cor) and TBS + 0.1% TWEEN-20 (TBST). Primary antibodies against rabbit-KIF18A (1:1,000; Bethyl Laboratories) and mouse-$\alpha$-tubulin (1:1,000; Sigma-Aldrich) were added in TBS overnight at 4°C. The membrane was washed with 2× TBST for 5 min each, then secondary antibodies (goat anti-rabbit IgG DyLight 800 conjugate and donkey anti-mouse IgG DyLight 680 conjugate, both at 1:10,000; Thermo

Fisher Scientific) added in 1:1 Odyssey Blocking Buffer and TBST for 1 h at room temperature. The membrane was washed with 2× TBST, then 1× TBS for 5 min each, and developed on an Odyssey CLx (Li-Cor).

## Microscopy

The cells were imaged on a Nikon Ti-E inverted microscope (Nikon Instruments) with Nikon objectives Plan Apo 40× DIC M N2 0.95 NA, Plan Apo $\lambda$ 60× 1.42NA, and APO 100× 1.49 NA with a Spectra-X light engine (Lumencore) and environment chamber. The following cameras were used: Clara cooled-CCD camera (Andor) and iXon X3 EMCCD camera (Andor). The Nikon Ti-E microscope is driven by Nikon Instruments Software (NIS)-Elements software (Nikon Instruments). TIRF microscopy was performed at room temperature using an inverted Eclipse Ti-E microscope (Nikon) with a 100× Apo TIRF objective lens (1.49 N.A.) and dual iXon Ultra Electron Multiplying CCD cameras, running NIS-Elements version 4.51.01. Rhodamine-labeled microtubules were excited with a 561 and a 590/50 filter. GFP-WT KIF18A$^{1–480}$ or GFP-sNL1 KIF18A$^{1–480}$ motors were excited with a 488 laser and a 525/50 filter. Alexa 647–labeled tau was excited with a 640 laser and 655 filter. Alexa 532–labeled rigor kinesin was excited with a 561 laser and a 590/50 filter. All movies were recorded with an acquisition time of 100 ms.

## KIF18A line scan analysis

Fixed and stained cells were imaged at 100× with 0.2 $\mu$m $z$-stacks taken through the full cell. Line scans were manually measured using the $\alpha$-tubulin and ACA fluorescent intensities to identify well-defined and locally isolated K-fibers. Lines were subdivided into central or peripheral spindle K-fiber classification based on their location in the spindle; ambiguous regions were avoided. The profile intensities for $\alpha$-tubulin, ACA, and GFP-KIF18A were measured and recorded. Each channel was normalized internally to its highest value. Line scans were aligned in block by peak ACA intensity and averaged for each pixel distance. SDs are reported. Statistical comparisons were performed by fitting a linear regression of the accumulation slope (–0.8 to 0 $\mu$m) and comparing each sNL variant to that of WT KIF18A at central and peripheral K-fibers using an F test. To determine the tip to lattice ratio of KIF18A fluorescence intensity, line scans along centrally located K-fibers were normalized and aligned to peak ACA intensity as described earlier, but then each line was sectioned between –0.768 and –0.32 $\mu$m (lattice) and –0.32 to 0.0 $\mu$m (tip) and the area under the curve calculated for each. These areas under the curve calculations were then divided to determine the ratio of tip to lattice KIF18A accumulation for each individual line scan, which was then graphed for each condition.

## Chromosome alignment analysis

Quantification of chromosome alignment was performed as described previously (Stumpff et al, 2012; Kim et al, 2014). Briefly, single focal planes were imaged of metaphase cells with both spindle poles in focus. The Plot Profile command in ImageJ was used to measure the distribution of ACA-labeled kinetochore fluorescence within a region of interest defined by the length of the spindle with a set height of 17.5 $\mu$m. The ACA signal intensity within the region of interest was averaged along each pixel column, normalized, and

plotted as a function of distance along the normalized spindle pole axis. These plots were analyzed by Gaussian fits using Igor Pro (WaveMetrics). The FWHM intensity for the Gaussian fit and the SL are reported for each cell analyzed.

### Taxol treatment of KIF18A sNL0–transfected cells

The cells were transfected with either control siRNA or KIF18A siRNA for 48 h, and after 24 h, the cells were transfected with GFP-KIF18A sNL0 following the transfection protocol described earlier. The cells were arrested in MG132 for 1–2 h, then 10 $\mu$M taxol was added 10 min before fixation. Fixation and immunofluorescence staining were followed as described earlier. Metaphase cells expressing GFP-KIF18A sNL0 were imaged and visually scored for any K-fiber end accumulation.

### Live cell imaging and mitotic timing analysis

Culturing medium was removed from transfected cells plated on poly-D-lysine–coated filming dishes (MatTek) and replaced with $CO_2$-independent imaging medium (Gibco) supplemented with 10% FBS and 1% penicillin and streptomycin. Using the 40× objective with differential interference contrast (DIC) and GFP fluorescent imaging, 10 randomly selected XY fields were programmed for imaging every 2 min for 16 h. For analysis, nuclear envelope breakdown and anaphase onset time points were determined for cells that completely divided and were detectably expressing GFP.

### KIF18A[1–480] protein purification

40 ng of GFP-KIF18A[1–480] WT-CFLAG or GFP-KIF18A[1–480] sNL1-CFLAG was transformed into 20 $\mu$l Rosetta cells (Novagen) and grown for 16 h at 37°C in 5 ml Terrific Broth (TB; Thermo Fisher Scientific) supplemented with 50 $\mu$g/ml kanamycin (Sigma-Aldrich). A starter culture was grown overnight at 37°C by adding 1 ml of the above into 25 ml TB with 50 $\mu$g/ml kanamycin for 16 h. The starter culture was then added to 500 ml TB with 50 $\mu$g/ml kanamycin and grown at 37°C until $OD_{600}$ reached 1.0–1.2. The cells were then chilled on ice for 5 min before inducing with 0.4 mM IPTG. The cultures were then grown at 30°C for 4 h, pelleted at 5,000$g$ for 20 min at 4°C, and the supernatant removed. The pellets were resuspended in one-fourth the volume of 1 g to 10 ml TSS buffer (50 mM Hepes, pH 7.0, 0.1 M KCl, 1 mM $MgCl_2$, and 10% glycerol) and frozen slowly at −80°C. The cells were thawed and then dropped over liquid nitrogen and ground into a fine powder. Then remaining three-fourth volume of TSS buffer was added, supplemented with the following: 1× Cytolitic B Cell Lysis Reagent (Sigma-Aldrich), 0.01 mM Mg-ATP, 1 mM PMSF, 1× protease and phosphatase inhibitor mini tablets, EDTA-free (Thermo Fisher Scientific), and 1 mg/ml DNAseI (Sigma-Aldrich). The cells were lysed using a Dounce homogenizer and then clarified at 23,000$g$ for 40 min at 4°C. The supernatant was flown over a FLAG column (ANTI-FLAG M2 Affinity Gel; Sigma-Aldrich) pre-equilibrated with TSS buffer supplemented with 0.01 mM Mg-ATP. The column was washed with 10 column volumes of TSS buffer with 0.01 mM Mg-ATP and then eluted with 0.1 mg/ml FLAG peptide (Sigma-Aldrich) in TSS buffer with 0.01 mM Mg-ATP. 1 ml of this elution buffer was added at a time, mixed with the affinity gel, incubated for 5 min, and then collected. Three to four fractions were collected, and fractions 2 and 3 were combined, divided into 50 $\mu$l

aliquots, and snap-frozen in liquid nitrogen. The samples were collected along the purification and prepared for gel electrophoresis by combining with 4× Laemmli buffer prepared with $\beta$ME (Bio-Rad). For analysis of protein preparation, fractions were detected with Western blot using antibodies against each end (GFP and FLAG). Briefly, the samples were run on precast 4–15% gradient gels (Bio-Rad), transferred to low-fluorescence polyvinylidene fluoride membrane (Bio-Rad), blocked for 1 h in 1:1 Odyssey Blocking Buffer (Li-Cor) and TBST. Primary antibodies against rabbit-GFP (1:1,000; Molecular Probes) and mouse-FLAG (1:1,000; Life Technologies) were added in 1:1 Odyssey Blocking Buffer and TBST overnight at 4°C. The membrane was washed with 2× TBST 5 min each and then secondary antibodies (goat anti-mouse IgG DyLight 680 conjugate and donkey anti-rabbit IgG DyLight 800 conjugate, both at 1:10,000; Thermo Fisher Scientific) added in 1:1 Odyssey Blocking Buffer and TBST for 1 h at room temperature. The membrane was washed with 2× TBST and then with 1× TBS for 5 min each and developed on an Odyssey CLx (Li-Cor).

### Single-molecule TIRF motility assays with and without obstacles

Bovine tubulin and tau (3RS) were purified and tau was labeled with Alexa 647 as reported previously (Stern et al, 2017). Rigor kinesin (*Rattus norvegicus* KIF5C[1–354] T93N) was purified from BL21 (DE3) *E. coli* and labeled with Alexa 532 as reported previously (Hoeprich et al, 2017). For motility assays, flow chambers used in in vitro TIRF experiments were constructed as previously described (Stern et al, 2017). Briefly, flow chambers were incubated with monoclonal anti-$\beta$ III (neuronal) antibodies (Sigma-Aldrich) at 33 $\mu$g/ml for 10 min and then washed twice with 0.5 mg/ml BSA (Sigma-Aldrich) and incubated for 2 min. 1 $\mu$M of microtubules (all experimental conditions) were administered and incubated for 10 min. Nonadherent microtubules were removed with a BRB80 wash (80 mM PIPES, 1 mM $MgCl_2$, 1 mM EGTA, 10 mM DTT, 1 mM $MgCl_2$, and an oxygen scavenger system [5.8 mg/ml glucose, 0.045 mg/ml catalase, and 0.067 mg/ml glucose oxidase; Sigma-Aldrich]) supplemented with 20 $\mu$M paclitaxel (Sigma-Aldrich) and 100 mM KCl. Kinesin motors in BRB80 + 100 mM KCl and 1 mM ATP were added to the flow cell just before image acquisition. For rhodamine-labeled microtubules, unlabeled GTP tubulin was mixed with rhodamine-labeled tubulin (Cytoskeleton) at a ratio of 100:1. The tubulin mixture was polymerized at 37°C for 20 min and stabilized with 20 $\mu$M paclitaxel in DMSO. For experiments containing tau, tubulin polymerization was performed as described earlier but in the absence of rhodamine-labeled tubulin. Polymerized microtubules were incubated with 25 nM Alexa 647–labeled tau (1:40 tau to tubulin ratio) or 80 nM Alexa 532 rigor kinesin (1:12.5 rigor to tubulin ratio) for an additional 20 min at 37°C. Tau/rigor and microtubules mixture was centrifuged at room temperature for 30 min at 21,130 $g$. The pellet was then resuspended in BRB80 + 100 mM KCl and 20 $\mu$M taxol solution.

### TIRF assay motility analysis

Motility events were analyzed as previously reported (Hoeprich et al, 2014, 2017). In brief, overall run length motility data were measured using the ImageJ (NIH) MTrackJ plug-in, for a frame-by-frame quantification of KIF18A motility. Characteristic and corrected run lengths and track lengths were calculated as previously described (Thompson et al, 2013). Average run length and velocity

were plotted as a histogram with mean and SD. Statistical significicance was determined by unpaired *t* test on corrected run lengths and velocities. Kymographs of motor motility were created using the MultipleKymograph ImageJ plug-in, with a line thickness of three.

### K-fiber density quantifications upon HURP or TACC3 knockdown or HURP overexpression

In NIS-elements, max z-projections were created from three 0.2 *μm* *z*-stacks on either side of the defined middle of cell (seven *z*-stacks in total). Then, circular regions of interest (~0.2 *μm* in area) were placed around individual kinetochores such that the end of the kinetochore intensity aligned with the edge of the circle most proximal to the center of the cell. Approximately 5–10 kinetochore measurements were taken per cell. Cytosolic background was determined by averaging two circles outside the spindle in each cell. Background subtracted *α*-tubulin fluorescent intensity was then normalized for each condition to the control treatment and graphed.

## Supplementary Information

## Acknowledgements

We wish to thank Puck Ohi for his thoughtful discussion and generous gift of the GFP-HURP plasmid created by Sophia Gayek, and Lynn Chrin for her technical assistance in purifying and labeling 3RS tau and rigor kinesin. We also thank Maria Kogan for writing a custom MATLAB program for line scan analysis. This work was funded by National Institutes of Health (NIH) R01GM121491 awarded to J Stumpff and NIH R01GM101066 to CL Berger. HLH Malaby was supported by a Department of Defense Peer-Reviewed Cancer Research Program Horizon Award (W81XWH-17-1-0371).

### Author Contributions

HLH Malaby: conceptualization, data curation, formal analysis, investigation, methodology, and writing—original draft, review, and editing.
DV Lessard: data curation, formal analysis, investigation, methodology, and writing—review and editing.
CL Berger: resources, supervision, and writing—review and editing.
J Stumpff: conceptualization, supervision, funding acquisition, methodology, and writing—original draft, review, and editing.

### Conflict of Interest Statement

The authors declare that they have no conflict of interest.

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
