## [Reviewer comments · Life Science Alliance]

Life Science Alliance

KIF18A's Neck Linker Permits Navigation of Microtubule Bound Obstacles within the Mitotic Spindle

Heidi Malaby, Dominique Lessard, Christopher Berger, and Jason Stumpff

Corresponding author(s): Jason Stumpff, University of Vermont

Review Timeline:

Submission Date:	2018-08-21
Editorial Decision:	2018-09-13
Revision Received:	2018-12-13
Editorial Decision:	2018-12-31
Revision Received:	2019-01-04
Accepted:	2019-01-07

Scientific Editor: Andrea Leibfried

Transaction Report:

DOI: 10.26508/lsa.201800169

September 13, 2018

Re: Life Science Alliance manuscript #LSA-2018-00169-T

Jason Stumpff
University of Vermont
Molecular Physiology and Biophysics
149 Beaumont Avenue
HSRF 118
Burlington, Vermont 05405

Dear Dr. Stumpff,

Thank you for submitting your manuscript entitled "The Human Kinesin Kif18A's Neck Linker Permits Navigation of Obstacles within the Mitotic Spindle" to Life Science Alliance. The manuscript was assessed by expert reviewers, whose comments are appended to this letter.

As you will see, all three reviewers appreciate your work and find it of value to others. The reviewers provide constructive input on how to further strengthen your work by performing a (relatively minor) revision. We would thus like to invite you to provide a revised version of your manuscript, following the reviewers' suggestions. We would be happy to discuss the individual revision points further with you should this be helpful.

Thank you for this interesting contribution to Life Science Alliance. We are looking forward to receiving your revised manuscript.

Sincerely,

Andrea Leibfried, PhD

Executive Editor
Life Science Alliance
Meyerhofstr. 1
69117 Heidelberg, Germany
t +49 6221 8891 502
e a.leibfried@life-science-alliance.org
www.life-science-alliance.org

- A letter addressing the reviewers' comments point by point.
- An editable version of the final text (.DOC or .DOCX) is needed for copyediting (no PDFs).
- High-resolution figure, supplementary figure and video files uploaded as individual files: See our detailed guidelines for preparing your production-ready images, <http://life-science-alliance.org/authorguide>
- Summary blurb (enter in submission system): A short text summarizing in a single sentence the study (max. 200 characters including spaces). This text is used in conjunction with the titles of papers, hence should be informative and complementary to the title and running title. It should describe the context and significance of the findings for a general readership; it should be written in the present tense and refer to the work in the third person. Author names should not be mentioned.

B. MANUSCRIPT ORGANIZATION AND FORMATTING:

Full guidelines are available on our Instructions for Authors page, <http://life-science-alliance.org/authorguide>

Reviewer #1 (Comments to the Authors (Required)):

The manuscript by Malaby et al examines the ability of the kinesin-8 motor KIF18A to navigate obstacles and regulate microtubule dynamics during chromosome congression. Building on previous work in the field, the authors examine the hypothesis that the 17 amino acid neck linker of KIF18A provides this motor with the ability to step around obstacles. To test this, they generate a series of truncations in the neck linker and examine the ability of the mutant motors to function in

chromosome congression using a knockdown-rescue approach and in their motility in single molecule assays. They find that mutant motors with shorter neck linkers results in less condensed chromatin at the metaphase plate in cells as well as an increased speed but decreased run length for single motors. This is the first study to examine the role of the neck linker not just in vitro but in a cellular context and is thus important for the field. The data are clear and rigorous and the manuscript is well-written. I recommend publication after addressing the following comments.

Major comments:

1. In the literature, the response of kinesins to obstacles seems to be very dependent on the kinesin and the obstacle. Since neither the rigor kinesin nor tau is an obstacle that KIF18A would encounter in the spindle (as far as I know), it would be nice to know if HURP is an obstacle for KIF18A and whether the neck linker helps KIF18A to navigate this obstacle. The effects in cells of HURP knockdown or overexpression are fairly minor although they are statistically significant.
2. In the case of KIF18A, the authors see no effect of adding rigor kinesin as an obstacle but a reduction in run-length when adding tau as an obstacle. The reduction in run length was more pronounced for the sNL1 construct in the presence of tau. Is the sNL1 construct also more sensitive to the rigor kinesin? Maybe there is an increase in pauses.
3. Given the influence of the KIF18A tail domain on processivity of this motor, it would be nice to see if the reduced run length upon shortening the neck linker is relevant in the context of the full length motor.

Minor comments:

4. The use of the truncated constructs to examine the effect of neck linker truncation on motility is appropriate but the designation of this construct as 480 is confusing. The 480 sounds like 480 amino acids were deleted when in reality this construct contains amino acids 1-480.
5. To make it clear that the experiments in cells were done under conditions to knockdown the endogenous protein by siRNA and not just overexpression of the KIF18A-GFP constructs, it would be helpful to include the label "siRNA KIF18A" along the y-axis of Figure 2A, the y-axis of Figure 3A, and the x-axis of Figure 3B.
6. Can the authors explain more about how/why they designed the neck linker truncations?
7. Please provide the averages for the graphs in Figure 5 in the top right corner of each graph.

Reviewer #2 (Comments to the Authors (Required)):

This paper from Malaby et al. looks at how Kif18A make its way along the crowded microtubule environment of the spindle. They present evidence that the neck linker gives Kif18A wiggle room to bypass crosslinkers. This is put forward as an argument for the distribution of the motor in mitotic spindles. I was happy to review this paper having already read it as a preprint on bioRxiv. Thank you to the authors for sharing their work with the community early. I have some comments below to help the authors improve their work.

There is a control missing in Figure 3. There is a statement "As expected, GFP alone did not rescue chromosome alignment, while GFP-WT Kif18A fully recovered alignment (Figure 3B)". To know this we need to see FWHM distance and SL distance for control RNAi cells with GFP expression.

Likewise, in Figure 4 the changes in NEB to AO timing are meaningless without a control.

Figure 1 is a bit minimalist and should be part of Figure 2.

p.4 "GFP-Kif18A sNL0 showed no K-fiber end accumulation and was unable to align chromosomes, suggesting that it may be inactive (Figure S1A-B)." This is the first experimental result described. Since Figure 2 has not been described, I had no idea what "K-fiber end accumulation" meant. Assuming the audience for this paper is bigger than the handful of people working on Kif18A, I think it would be better to describe the wild-type distribution before getting stuck into what one of the mutants looks like.

Figs 2, 3 and 6 have scale bars with the label 2 μm . Sorry to be picky but it's my role! A few problems here: μm is not the correct SI unit, I personally don't like to see the value written above the bar, and if all the bars are the same they're not needed in every panel. I also think 2 microns isn't meaningful as a scale bar. 10 microns (approx width/length of spindle is more useful).

Fig 3B it's not clear what is plotted for bars in the scatter plots. Mean or median {plus minus} what? The legend mentions 95% CI but from the way it's written this may refer to the alpha level for the ANOVA and not the error bar.

Fig 4 legend missing a detail on how the p-values were calculated. Is it a comparison of time in mitosis using n =individual cells or were the shapes of the distributions compared using K-S test for $n=3$?

Fig 6C the font is different for lattice/tip vs all other figures

Reviewer #3 (Comments to the Authors (Required)):

In this manuscript, Malaby et al. describe a function of the neck linker of Kif18A in the accumulation of this kinesin on microtubule plus-ends. Previously, neck-linker functions of other kinesins have been investigated in in vitro systems by modifying the neck-linker length, showing that this affects the run-length and processivity of the kinesin, but also its ability to navigate around obstacles. However, the role of the kinesin neck-linker flexibility in cells was not yet understood. Here, Malaby and colleagues show, by using short neck-linker variants of Kif18A, that the length of the linker is important to enable passing of HURP on central K-fibers in order for Kif18A to accumulate on microtubule plus-ends.

A physiological role of kinesin side-stepping has not been shown in vivo before. Even though the topic is quite specialized, the writing of the manuscript made the topic very accessible. The work provided in this manuscript seems of sufficient quality and supports the idea of the need of a longer neck-linker for navigating on protein-dense microtubules. However, where the title suggests a broadly applicable phenotype, the manuscript and the in vivo experiments seem to narrow it down to a HURP-Kif18A specific interaction, and only on central K-fibers. Where some experiments are elegant, others seem to lack important controls or were over interpreted. Therefore, I feel the

authors should do some additional experiments, and add appropriate controls or quantifications as suggested below. If the authors are able to answer the points listed below, I would advise to consider this manuscript for publication in Life Science Alliance.

Major points:

- The manuscript is very much focused on HURP as an obstacle for Kif18A. However, as also pointed out by the authors, the spindle is a very protein dense environment with many additional (large) MAP's. To show that the data is not only applicable to HURP but also to other MAP's, would it be possible to show the same effects by over-expressing and/or down-regulating other microtubule bound proteins? If the authors could provide additional data with other microtubule obstacles it would greatly enhance the importance of this work, as it would suggest a more general role of the neck-linker in navigating around protein-dense microtubules. If not, I would suggest to down-tune the title of the manuscript.
- During the course of the paper, it becomes apparent that the sNL mutants behave differently on central versus peripheral microtubules, but no separate measurements are shown. As the behavior seems to be so different on these different types of MT's, it would be very interesting and important to make the distinction between central and peripheral microtubules by measuring central and peripheral microtubules separately. This would apply to figures 2, 3 and 6.
- In the discussion, page 10, it would be nice to elaborate a bit more on the statement that a decrease in run-length does not fully explain the localization phenotype on peripheral MT's. The data provided in Figure S5 is more an observation, rather than an explanation. Besides, the data in Figure S5 showing that even at peripheral MT's, HURP and Kif18A are mutually exclusive, is rather incomplete. Only one line-scan is provided and a more comprehensive quantification would be recommended where both peripheral and central microtubules are quantified in multiple cells. Especially since the mutants localize there, while peripheral MT's are longer than central ones. What would the hypothesis be? Do less proteins localize on peripheral MT's?
- I have several problems with the data shown in Figure 6:
 - o In 6A, when comparing the representative images of sNL1 and sNL1 + siHURP, I am not able to see the increase in central sNL1 localization upon HURP depletion.
 - o In 6C, it is unclear how the data is normalized exactly. It is odd to me why the GFP-sNL1 does not show a different ratio as compared to the GFP-WT but maybe the normalization methods will shed light on this issue. In line with this, absolute numbers would make these graphs better interpretable.
 - o To exclude that the observed differences in Figure 6C can be attributed to changes in MT density as a consequence of the inflicted treatments, it would be valuable to have a similar graph shown for tubulin intensity (ratio tip/lattice).
 - o In 6C, according to their previous data, WT-Kif18A is able to navigate around obstacles. Why does overexpression of HURP then affect WT-Kif18A accumulation?
 - o Why does overexpression of HURP not give any effect on the distribution of sNL1? Even if a discrimination between central and peripheral MT's would be made, would there still be no effect? Please clarify.

Minor points:

- The authors describe a difference of sNL localization on central and peripheral microtubules. However, they make use of the term central spindle on many occasions. As the central spindle is a different structure that classically refers to anaphase structure of the anti-parallel microtubule overlap, it would be less confusing to use terms like central microtubules.
- Indicate the use of siKif18A in Fig. 2. Besides, a western-blot showing knock-down efficiency when expressing the different mutants would be recommended.

- An alternative explanation for the defects seen with sNL0 could be that it is actually active but it is not able to navigate past anything at all and therefore displays the strongest phenotype. Was the motor activity of this mutant established?
- Figure 3A: The authors mention that GFP alone does not rescue the phenotype, however, one can only conclude that if the untransfected condition is shown. An alternative is to rephrase this sentence.
- Figure 5: add the values of table S1 to the figure (like in figure S3), this will make the effects observed with sNL1 clearer as they are depicted in separate graphs.
- I wonder why the authors do not show a sNL mutant in figure S3? The Rigor kinesin is considered an even stronger roadblock so possibly even larger effects could be observed.
- Page 6: sentence says 'and Tau', should be 'or Tau'.
- All assays are done with exogenous GFP-Kif18A constructs. This is a weakness of the story, however, I do realize that making endogenous mutants would be a challenging approach. To at least minimize variation induced by expression levels, it is important to know to which levels the GFP constructs are expressed. The current graphs show normalized data. It is unclear to me how the normalization was exactly done and it would be informative to provide evidence of the average expression levels in absolute values. An additional western blot would provide additional evidence for comparable expression levels. Transient transfection of a construct under a CMV promoter is expected to give quite some cell-to-cell variation so this is an important point to address.

Malaby et al.

Response to Reviewers:

We would like to thank the reviewers for taking the time to evaluate our manuscript. The paper has been significantly improved by their suggestions, and for that we are sincerely grateful.

Our responses to each of the reviewers' comments are described in detail below.

Reviewer #1:

The manuscript by Malaby et al examines the ability of the kinesin-8 motor KIF18A to navigate obstacles and regulate microtubule dynamics during chromosome congression. Building on previous work in the field, the authors examine the hypothesis that the 17 amino acid neck linker of KIF18A provides this motor with the ability to step around obstacles. To test this, they generate a series of truncations in the neck linker and examine the ability of the mutant motors to function in chromosome congression using a knockdown-rescue approach and in their motility in single molecule assays. They find that mutant motors with shorter neck linkers results in less condensed chromatin at the metaphase plate in cells as well as an increased speed but decreased run length for single motors. This is the first study to examine the role of the neck linker not just in vitro but in a cellular context and is thus important for the field. The data are clear and rigorous and the manuscript is well-written. I recommend publication after addressing the following comments.

Major comments:

1. In the literature, the response of kinesins to obstacles seems to be very dependent on the kinesin and the obstacle. Since neither the rigor kinesin nor tau is an obstacle that KIF18A would encounter in the spindle (as far as I know), it would be nice to know if HURP is an obstacle for KIF18A and whether the neck linker helps KIF18A to navigate this obstacle. The effects in cells of HURP knockdown or overexpression are fairly minor although they are statistically significant.

We agree that determining whether and how HURP directly affects KIF18A motility is an important question. We attempted to address this using the N-terminal region of HURP, which disrupts KIF18A localization in cells when overexpressed (PMID: 21924616). While recombinant HURP has been expressed in bacteria for structural studies, we were unable to find conditions that permitted purification of soluble HURP with consistent behavior on microtubules in single molecule assays. As the reviewer points out, there are caveats to the approach of using tau as a microtubule obstacle for KIF18A, however, these experiments allowed unambiguous conclusions about KIF18A neck linker function. In our revision, we have also included new data showing that depletion of TACC3, a component of the K-fiber mesh that binds and connects kinetochore microtubules together (PMID: 26090906), increases plus-end localization of the sNL1 mutant (Figure S5). This finding supports our hypothesis that KIF18A's neck linker permits navigation of microtubule-bound obstacles on K-fibers and indicates the effects of HURP are not specific.

2. In the case of KIF18A, the authors see no effect of adding rigor kinesin as an obstacle but a reduction in run-length when adding tau as an obstacle. The reduction in run length was more pronounced for the sNL1 construct in the presence of tau. Is the sNL1 construct also more sensitive to the rigor kinesin? Maybe there is an increase in pauses.

We have included new data in the revision from single molecule measurements of WT and sNL1 run lengths and velocities in the presence or absence of rigor kinesin (Figure S3). The presence of rigor kinesin reduces the sNL1 mutant run length 23% compared to 12% for the WT motor (Figure S3 B-D). This trend is consistent with the effects of tau on WT and sNL1 run-lengths (Figure 4). We believe the difference in the magnitudes of the effects measured in the presence of tau and rigor kinesin is likely explained by differences in the relative density of the two obstacles on microtubules in these assays (e.g. compare images in Figure 4E and Figure S3C). This interpretation is supported by our previous work with kinesin-2 motors, whose run lengths are also dependent on the length of the neck linker and obstacle density (PMID: 28267259).

3. Given the influence of the KIF18A tail domain on processivity of this motor, it would be nice to see if the reduced run length upon shortening the neck linker is relevant in the context of the full length motor.

We agree with the reviewer that determining how KIF18A's neck linker length and the non-motor microtubule-binding activity of its tail combine to affect run-length is an interesting question. We attempted to address this using *in vitro* motility assays with full-length GFP-WT KIF18A and GFP-sNL1 expressed in mammalian cell extracts. This approach has been used successfully to test physiologically relevant versions of kinesins by us and others (bioRxiv doi: <https://doi.org/10.1101/410860>; PMIDs: 24850887, 24706892, and 26018074). In previous studies, motor concentrations were high enough to permit dilution of extracts 100-500 fold, greatly reducing other cytosolic components added to the motility assay. Unfortunately, we found that this technique was not as effective for KIF18A, perhaps because it is nuclear localized in interphase and/or its expression is tightly controlled throughout the cell cycle. In order to detect KIF18A motility events, we were only able to dilute the extracts ~10-fold or less. This caused issues with the motility assay itself and is expected to introduce significant amounts of other cytosolic components, including microtubule-associated proteins. Thus, expression and purification of WT and sNL1 proteins via a baculovirus system will be required to unambiguously test the combined influence of the tail and neck linker domains on KIF18A's processivity. This approach would require a much larger time investment than was allotted for revision of this manuscript. Furthermore, our cellular data indicate that the full-length sNL1 mutant is capable of reaching the ends of peripheral kinetochore microtubules and central kinetochore microtubules when putative microtubule obstacles are reduced. We interpret these results to mean that the run-length of the full-length mutant is sufficiently long enough to target it to kinetochore-microtubule ends under these conditions. Thus, we do not feel that additional measurements of processivity *in vitro* will significantly alter our interpretation of KIF18A's neck linker function during mitosis.

Malaby et al.

Minor comments:

4. The use of the truncated constructs to examine the effect of neck linker truncation on motility is appropriate but the designation of this construct as 480 is confusing. The 480 sounds like 480 amino acids were deleted when in reality this construct contains amino acids 1-480.

We thank the reviewer for this suggestion. To avoid confusion, we have changed the nomenclature for the C-terminal truncated KIF18A constructs to KIF18A¹⁻⁴⁸⁰ in the revised manuscript.

5. To make it clear that the experiments in cells were done under conditions to knockdown the endogenous protein by siRNA and not just overexpression of the KIF18A-GFP constructs, it would be helpful to include the label "siRNA KIF18A" along the y-axis of Figure 2A, the y-axis of Figure 3A, and the x-axis of Figure 3B.

We appreciate this suggestion and have added labels to all relevant figures, including Figure 1, Figure 2, Figure 3, Figure 5, Figure S1, Figure S2, and Figure S5.

6. Can the authors explain more about how/why they designed the neck linker truncations?

The neck linker truncations were designed using coiled-coil prediction programs and sequence alignments as done previously for kinesin-1 and kinesin-2 (PMID: 20471270). Because more recent work showed that these prediction methods can be inaccurate in determining the start of a kinesin's coiled-coil region (PMID: 27462072), we empirically tested 4 different KIF18A mutants that each contained 3 amino acid truncations within the predicted neck linker region. We reasoned that this approach would allow us to differentiate between phenotypes caused by shortening the neck linker from those due to deletion of a particularly critical amino acid. Consistent with this logic, three of these mutants (sNL1, sNL2, and sNL3) behaved similarly in mitotic cells, while the fourth (sNL0) displayed localization defects similar to those observed for mutations that inactivate KIF18A. Since sNL0 removes the C-terminal end of the predicted neck linker, that deletion may affect the formation of the coiled-coil region. This information is included in the revised manuscript (page 4).

7. Please provide the averages for the graphs in Figure 5 in the top right corner of each graph.

We have added this information to Figure 5 (which is now Figure 4) in the revised manuscript.

Reviewer #2:

Malaby et al.

This paper from Malaby et al. looks at how Kif18A make its way along the crowded microtubule environment of the spindle. They present evidence that the neck linker gives Kif18A wiggle room to bypass crosslinkers. This is put forward as an argument for the distribution of the motor in mitotic spindles. I was happy to review this paper having already read it as a preprint on bioRxiv. Thank you to the authors for sharing their work with the community early. I have some comments below to help the authors improve their work.

There is a control missing in Figure 3. There is a statement "As expected, GFP alone did not rescue chromosome alignment, while GFP-WT Kif18A fully recovered alignment (Figure 3B)". To know this we need to see FWHM distance and SL distance for control RNAi cells with GFP expression.

This is a good point. In fact, we have previously shown that transient transfection of GFP-KIF18A significantly improves chromosome alignment in KIF18A KD cells but does not fully restore alignment to control cell levels (PMID: 25208566). This is likely due to cell-to-cell variations in rescue construct expression. We have edited the description of these results in the revised manuscript to be more precise (page 5).

Likewise, in Figure 4 the changes in NEB to AO timing are meaningless without a control.

Again, we agree that it is not possible to determine from our data whether mitotic timing is restored to control levels without making these measurements in a control siRNA treated sample. However, we do not agree that the data are meaningless without this condition. The experiment in Figure 4 (now Figure 3) includes both a positive (GFP-KIF18A WT) and negative control (GFP alone) for the effects of GFP-KIF18A-sNL1 on mitotic timing. Furthermore, our data are consistent with a previous study that showed the time from NEB to anaphase in HeLa cells, which is normally ~40 min, is increased to ~180 min after KIF18A KD and restored to ~40 min when GFP-KIF18A is expressed (PMID: 25048371). We have edited our description of this experiment in the revised manuscript (now Figure 3) to more accurately reflect the data (page 5).

Figure 1 is a bit minimalist and should be part of Figure 2.

We appreciate this suggestion and have combined Figures 1 and 2 in the revised manuscript.

p.4 "GFP-Kif18A sNL0 showed no K-fiber end accumulation and was unable to align chromosomes, suggesting that it may be inactive (Figure S1A-B)." This is the first experimental result described. Since Figure 2 has not been described, I had no idea what "K-fiber end accumulation" meant. Assuming the audience for this paper is bigger than the handful of people working on Kif18A, I think it would be better to describe the wild-type distribution before getting stuck into what one of the mutants looks like.

We thank the reviewer for this suggestion. We have attempted to present the sNL0 data with a more logical flow in the revised manuscript (pages 4-5).

Malaby et al.

Figs 2, 3 and 6 have scale bars with the label 2 μm . Sorry to be picky but it's my role! A few problems here: μm is not the correct SI unit, I personally don't like to see the value written above the bar, and if all the bars are the same they're not needed in every panel. I also think 2 microns isn't meaningful as a scale bar. 10 microns (approx width/length of spindle is more useful).

We appreciate these suggestions. We have changed “ μm ” to “ μm ” throughout the revised manuscript and have removed excess scale bars. We do prefer the label in the figure to facilitate interpretation of the data without a need to read the figure legend.

Fig 3B it's not clear what is plotted for bars in the scatter plots. Mean or median {plus minus} what? The legend mentions 95% CI but from the way it's written this may refer to the alpha level for the ANOVA and not the error bar.

We apologize for the confusion and have edited the figure legend (now Figure 2) to clarify that the mean \pm standard deviation are shown. As the reviewer suggests, the 95% confidence interval refers to the ANOVA test.

Fig 4 legend missing a detail on how the p-values were calculated. Is it a comparison of time in mitosis using n =individual cells or were the shapes of the distributions compared using K-S test for $n=3$?

We have clarified in the revised figure legend (now Figure 3) that the p-values were calculated from a comparison of time in mitosis using n = individual cells.

Fig 6C the font is different for lattice/tip vs all other figures

We thank the reviewer for catching this error, it has been fixed in the revised version.

Reviewer #3:

In this manuscript, Malaby et al. describe a function of the neck linker of Kif18A in the accumulation of this kinesin on microtubule plus-ends. Previously, neck-linker functions of other kinesins have been investigated in vitro systems by modifying the neck-linker length, showing that this affects the run-length and processivity of the kinesin, but also its ability to navigate around obstacles. However, the role of the kinesin neck-linker flexibility in cells was not yet understood. Here, Malaby and colleagues show, by using short neck-linker variants of Kif18A, that the length of the linker is important to enable passing of HURP on central K-fibers in order for Kif18A to accumulate on microtubule plus-ends.

A physiological role of kinesin side-stepping has not been shown in vivo before. Even though the topic is quite specialized, the writing of the manuscript made the topic very accessible. The work provided in this manuscript seems of sufficient quality and supports the idea of the need of a longer neck-linker for navigating on protein-dense microtubules. However, where the title suggests a broadly applicable phenotype, the manuscript and

Malaby et al.

the in vivo experiments seem to narrow it down to a HURP-Kif18A specific interaction, and only on central K-fibers. Where some experiments are elegant, others seem to lack important controls or were over interpreted. Therefore, I feel the authors should do some additional experiments, and add appropriate controls or quantifications as suggested below. If the authors are able to answer the points listed below, I would advise to consider this manuscript for publication in Life Science Alliance.

Major points:

- The manuscript is very much focused on HURP as an obstacle for Kif18A. However, as also pointed out by the authors, the spindle is a very protein dense environment with many additional (large) MAP's. To show that the data is not only applicable to HURP but also to other MAP's, would it be possible to show the same effects by over-expressing and/or down-regulating other microtubule bound proteins? If the authors could provide additional data with other microtubule obstacles it would greatly enhance the importance of this work, as it would suggest a more general role of the neck-linker in navigating around protein-dense microtubules. If not, I would suggest to down-tune the title of the manuscript.

We thank the reviewer for this suggestion. In the revised manuscript, we have included new data (Figure S5) indicating that knockdown of TACC3, another MAP involved in spindle assembly and maintenance, leads to similar effects on the localization of KIF18A WT and sNL1 to those observed following HURP knockdown. These data suggest that the effects of HURP as a putative obstacle for the targeting of KIF18A to K-fiber ends can be extended to other K-fiber-associated MAPs.

- During the course of the paper, it becomes apparent that the sNL mutants behave differently on central versus peripheral microtubules, but no separate measurements are shown. As the behavior seems to be so different on these different types of MT's, it would be very interesting and important to make the distinction between central and peripheral microtubules by measuring central and peripheral microtubules separately. This would apply to figures 2, 3 and 6.

We apologize for the difficulty in following which subset of K-fibers were analyzed in each experiment. We have edited the text for clarification. Figure 1 displays images from a central slice of the spindle, and the quantifications shown are from central MTs. To complement these analyses, Figure S1 shows images taken of the edges of spindles under each experimental condition, and the quantification of GFP-KIF18A constructs in this figure were made on peripheral K-fibers. The quantification of chromosome alignment in Figure 2 required images through the center of the spindle where both spindle poles were in focus. Thus, both central and peripheral kinetochores were measured. KIF18A localization in cells with altered HURP (Figure 5) and TACC3 (Figure S5) levels was quantified along K-fibers at the center of the spindle.

- In the discussion, page 10, it would be nice to elaborate a bit more on the statement that a decrease in run-length does not fully explain the localization phenotype on peripheral

Malaby et al.

MT's. The data provided in Figure S5 is more an observation, rather than an explanation. Besides, the data in Figure S5 showing that even at peripheral MT's, HURP and Kif18A are mutually exclusive, is rather incomplete. Only one line-scan is provided and a more comprehensive quantification would be recommended where both peripheral and central microtubules are quantified in multiple cells. Especially since the mutants localize there, while peripheral MT's are longer than central ones. What would the hypothesis be? Do less proteins localize on peripheral MT's?

We have clarified our statement that the localization of sNL mutants in mitotic cells is unlikely due simply to a reduced run length. A general reduction in run length would be expected to limit the accumulation of KIF18A at all K-fiber ends. In fact, one might predict the effect to be more prominent on the longer microtubules predicted to be in peripheral K-fibers. Our data indicates the opposite scenario occurs. KIF18A sNL mutants fail to accumulate on K-fibers at the center of the spindle but do localize to the ends of peripheral K-fibers.

A possible explanation for this result is suggested by the observation that the peaks of KIF18A and HURP fluorescence appear to be spatially separated near the ends of peripheral K-fibers. We have clarified in the revised manuscript that the lines cans displayed are representative of many line scans obtained from multiple cells expressing each construct (see legend for Figure 6). We have also included the measurements from all line scans as source data. These data are consistent with the working model presented in Figure 6.

- I have several problems with the data shown in Figure 6:
 - o In 6A, when comparing the representative images of sNL1 and sNL1 + siHURP, I am not able to see the increase in central sNL1 localization upon HURP depletion.

The example images displayed in the figure (now Figure 5) are truly representative, rather than the brightest cells we imaged, and the expression level of each construct is comparable among the cells shown. To improve visualization of sNL1 localization at K-fiber ends, we have included a new panel of zoomed-in images with arrows to indicate GFP-KIF18A foci near kinetochores.

- o In 6C, it is unclear how the data is normalized exactly. It is odd to me why the GFP-sNL1 does not show a different ratio as compared to the GFP-WT but maybe the normalization methods will shed light on this issue. In line with this, absolute numbers would make these graphs better interpretable.

This is an astute observation. For the ratio of tip to lattice accumulation the data were normalized to one for each KIF18A variant under the control condition (KIF18A knockdown only). Due to the reviewer's question, we realized that this normalization was unnecessary and somewhat obscured the data. In the revised manuscript, we have plotted the raw ratios of tip to lattice accumulation (now in Figure 5). We believe this new representation is easier to follow, and we thank the reviewer for bringing this to our attention.

Malaby et al.

o To exclude that the observed differences in Figure 6C can be attributed to changes in MT density as a consequence of the inflicted treatments, it would be valuable to have a similar graph shown for tubulin intensity (ratio tip/lattice).

Altering HURP expression in cells does affect K-fiber density (PMID: 18321990). We have quantified this ourselves in Figure S4 by measuring the average intensity of MTs adjacent to kinetochores. The effects of microtubule density changes on KIF18A localization under these conditions are also discussed in the text (pages 7-8).

o In 6C, according to their previous data, WT-Kif18A is able to navigate around obstacles. Why does overexpression of HURP then affect WT-Kif18A accumulation?

This is a good question. Our interpretation of these data is that HURP overexpression increases obstacle density enough to limit WT-KIF18A motility. This interpretation agrees with our previous measurements of kinesin-2 motility in the presence of different obstacle densities in vitro (PMID: 28267259). Kinesin-2 motors also possess an extended neck linker that permits navigation around obstacles (PMID: 24739168). However, at high obstacle density kinesin-2 run length is decreased, suggesting a limitation on the “agility” afforded by increased neck linker length (PMID: 28267259).

o Why does overexpression of HURP not give any effect on the distribution of sNL1? Even if a discrimination between central and peripheral MT's would be made, would there still be no effect? Please clarify.

The sNL1 mutant does not accumulate detectably on central K-fibers under normal conditions, thus we would not expect that adding additional obstacles would change the localization there. These are the data presented in Figure 5. We agree with the reviewer that accumulation at peripheral K-fibers also seems to be unchanged (we also quantified this in one data set to confirm) and that this is somewhat surprising. However, these data are consistent with our observation that KIF18A and HURP are spatially separated near the ends of individual K-fibers (Figure 6A). Perhaps HURP does not localize to certain microtubule subsets within these K-fibers regardless of expression level, which in turn allows modest accumulation of the sNL1 mutant.

Minor points:

- The authors describe a difference of sNL localization on central and peripheral microtubules. However, they make use of the term central spindle on many occasions. As the central spindle is a different structure that classically refers to anaphase structure of the anti-parallel microtubule overlap, it would be less confusing to use terms like central microtubules.

We apologize for this confusion and have removed the term “central spindle” from the revised manuscript.

Malaby et al.

- Indicate the use of siKif18A in Fig. 2. Besides, a western-blot showing knock-down efficiency when expressing the different mutants would be recommended.

We have updated all figures to indicate where KIF18A siRNA treated cells were analyzed. We have also included measurements of knockdown efficiency via Western blots in cells expressing KIF18A variants (Figure 2).

- An alternative explanation for the defects seen with sNLO could be that it is actually active but it is not able to navigate past anything at all and therefore displays the strongest phenotype. Was the motor activity of this mutant established?

We suspect that the sNLO mutant has a coiled-coil defect due to the removal of H369. However, the reviewer is correct in suggesting that there could be other explanations for the phenotypes observed in cells expressing this mutant. We have backed-off on our conclusion that the sNLO mutant is inactive since we have not directly tested this. The fact that sNLO does not behave like the other three mutants with shortened neck linkers suggests its localization and functional defects are not solely due to a reduced neck linker length. Thus, further analyses of this mutant would not be directly related to the goals of the present study.

- Figure 3A: The authors mention that GFP alone does not rescue the phenotype, however, one can only conclude that if the untransfected condition is shown. An alternative is to rephrase this sentence.

We have edited the description of these results in the revised manuscript to be more precise (page 5).

- Figure 5: add the values of table S1 to the figure (like in figure S3), this will make the effects observed with sNL1 clearer as they are depicted in separate graphs.

We have added the average run length and velocity values to the corresponding graphs in the figure (now Figure 4).

- I wonder why the authors do not show a sNL mutant in figure S3? The Rigor kinesin is considered an even stronger roadblock so possibly even larger effects could be observed.

We have added measurements of sNL1 motility in the presence of rigor kinesin to the revised manuscript (Figure S3). Please see Reviewer 1, point #2 for a full description.

- Page 6: sentence says 'and Tau', should be 'or Tau'.

We have fixed this typo.

- All assays are done with exogenous GFP-Kif18A constructs. This is a weakness of the story, however, I do realize that making endogenous mutants would be a challenging approach. To at least minimize variation induced by expression levels, it is important to

Malaby et al.

know to which levels the GFP constructs are expressed. The current graphs show normalized data. It is unclear to me how the normalization was exactly done and it would be informative to provide evidence of the average expression levels in absolute values. An additional western blot would provide additional evidence for comparable expression levels. Transient transfection of a construct under a CMV promotor is expected to give quite some cell-to-cell variation so this is an important point to address.

We appreciate this concern. To address it, we have included quantified western blots of cells lysed at the same time points following transfection as all fixed cells. These data have been added to Figure 1. We find that 1) endogenous KIF18A protein remains knocked-down with high efficiency following transfection of exogenous GFP-KIF18A, and 2) there is no significant variation amongst the expression of GFP-KIF18A constructs.

December 31, 2018

RE: Life Science Alliance Manuscript #LSA-2018-00169-TR

Dr. Jason Stumpff
University of Vermont
Molecular Physiology and Biophysics
149 Beaumont Avenue
HSRF 118
Burlington, Vermont 05405

Dear Dr. Stumpff,

Thank you for submitting your revised manuscript entitled "KIF18A's Neck Linker Permits Navigation of Microtubule Bound Obstacles within the Mitotic Spindle". As you will see, reviewer #1 appreciates the introduced changes and thinks that you have adequately addressed all concerns of the reviewers. We would thus be happy to publish your paper in Life Science Alliance pending final revisions necessary to meet our formatting guidelines:

- please mention the arrows shown in fig5A and B and in FigS5A in the figure legends.
- please link you profile in our submission system to your ORCID ID, you should have received an email with instructions on how to do so.

A. FINAL FILES:

-- High-resolution figure, supplementary figure and video files uploaded as individual files: See our detailed guidelines for preparing your production-ready images, <http://life-science-alliance.org/authorguide>

B. MANUSCRIPT ORGANIZATION AND FORMATTING:

Full guidelines are available on our Instructions for Authors page, <http://life-science-alliance.org/authorguide>

Sincerely,

Reviewer #1 (Comments to the Authors (Required)):

The revised manuscript by Malaby et al provides important new information about how the kinesin-8 motor KIF18A navigates the mitotic spindle to accumulate at plus ends of microtubules and impact chromosome congression. The data are rigorous and clear, the manuscript is well-written,

the revisions have clarified the previous concerns, and I recommend publication of this important work.

January 7, 2019

RE: Life Science Alliance Manuscript #LSA-2018-00169-TRR

Dr. Jason Stumpff
University of Vermont
Molecular Physiology and Biophysics
149 Beaumont Avenue
HSRF 118
Burlington, Vermont 05405

Dear Dr. Stumpff,

Thank you for submitting your Research Article entitled "KIF18A's Neck Linker Permits Navigation of Microtubule Bound Obstacles within the Mitotic Spindle". It is a pleasure to let you know that your manuscript is now accepted for publication in Life Science Alliance. Congratulations on this interesting work.

DISTRIBUTION OF MATERIALS:

Again, congratulations on a very nice paper. I hope you found the review process to be constructive and are pleased with how the manuscript was handled editorially. We look forward to future exciting submissions from your lab.

Sincerely,
